# Association of Mediterranean Diet Scores with Psychological Distress in Pregnancy: The Japan Environment and Children’s Study

**DOI:** 10.3390/nu17233697

**Published:** 2025-11-25

**Authors:** Yuri Takahashi, Zen Watanabe, Noriyuki Iwama, Natsumi Kumagai, Hirotaka Hamada, Hikaru Karumai-Mori, Seiya Izumi, Emi Yokoyama, Yasuno Takahashi, Takeki Sato, Jumpei Toratani, Kazuma Tagami, Hasumi Tomita, Masahito Tachibana, Mami Ishikuro, Taku Obara, Hirohito Metoki, Tomohisa Suzuki, Yuichiro Miura, Chiharu Ota, Shinichi Kuriyama, Takahiro Arima, Nobuo Yaegashi, Masatoshi Saito

**Affiliations:** 1Department of Obstetrics and Gynecology, Tohoku University Graduate School of Medicine, 1-1 Seiryomachi, Sendai 980-8574, Miyagi, Japan; takahashi.yuri.p7@dc.tohoku.ac.jp (Y.T.);; 2Center for Maternal and Perinatal Medicine, Tohoku University Hospital, 1-1 Seiryomachi, Sendai 980-8574, Miyagi, Japan; 3Division of Molecular Epidemiology, Department of Preventive Medicine and Epidemiology, Tohoku Medical Megabank Organization, Tohoku University, 2-1 Seiryomachi, Sendai 980-8573, Miyagi, Japan; 4Division of Molecular Epidemiology, Tohoku University Graduate School of Medicine, 2-1 Seiryomachi, Sendai 980-8575, Miyagi, Japan; 5Division of Public Health, Hygiene and Epidemiology, Tohoku Medical Pharmaceutical University, 1-15-1 Fukumuro, Sendai 983-8536, Miyagi, Japan; 6Tohoku Medical Megabank Organization, Tohoku University, 2-1 Seiryomachi, Sendai 980-8573, Miyagi, Japan; 7Environment and Genome Research Center, Tohoku University Graduate School of Medicine, 2-1 Seiryomachi, Sendai 980-8575, Miyagi, Japan; 8Department of Paediatrics, Tohoku University Graduate School of Medicine, 1-1 Seiryomachi, Sendai 980-8574, Miyagi, Japan; 9International Research Institute of Disaster Science, Tohoku University, 468-1 Aoba, Aramaki, Sendai 980-8572, Miyagi, Japan; 10Department of Maternal and Fetal Therapeutics, Tohoku University Graduate School of Medicine, 1-1 Seiryomachi, Sendai 980-8574, Miyagi, Japan

**Keywords:** mediterranean diet, MDS (mediterranean diet score), rMED (relative mediterranean diet), PMDS (mediterranean diet score in pregnancy), perinatal mental disorders, pregnancy, psychological distress

## Abstract

**Background/Objectives:** Perinatal mental disorders are important health issues that affect both mothers and their children. The Mediterranean diet (MD) is one of the most well-recognized healthy dietary patterns worldwide. Recent evidence suggests that MD may prevent or reduce the risk of perinatal mental disorders. This study investigated the association between MD adherence during pregnancy and psychological distress in a large cohort of Japanese births. **Methods:** Data were obtained from 80,271 pregnant women who participated in the Japan Environment and Children’s Study. Adherence to the MD was assessed using three scoring methods: the Mediterranean Diet Score (MDS), relative Mediterranean Diet (rMED), and Mediterranean Diet Score for Pregnancy (PMDS). Psychological distress was defined as a score of ≥13 on the Kessler 6-item Psychological Distress Scale (K6). Modified Poisson regression models were used to estimate risk ratios (RRs) and 95% confidence intervals (CIs). Population attributable fractions (PAFs) were calculated to quantify the proportion of psychological distress attributable to low MD adherence. **Results:** The adjusted RR for psychological distress in the low PMDS group was 1.17 (95% CI: 1.07, 1.28) compared with the high PMDS group. The adjusted PAF for low PMDS was 10.43% (95% CI: 4.81, 16.06). No statistically significant associations were detected between psychological distress and the MDS or rMED scores. **Conclusions:** This study suggests that high adherence to MD based on the PMDS during pregnancy may reduce the risk of psychological distress in Japanese women. Therefore, promoting MD during pregnancy could be a potential strategy for preventing perinatal mental disorders.

## 1. Introduction

Perinatal mental disorders are critical health concerns affecting both mothers and their children. Depression and anxiety disorders during pregnancy have been associated with negative outcomes, including preterm delivery [1,2,3,4], low birth weight [5,6], and preeclampsia [7]. Specifically, pregnant women who screened positive for depressive symptoms (Edinburgh Postnatal Depression Scale [EPDS] ≥12 or thoughts of self-harm) at 24–28 weeks of gestation had an adjusted odds ratio (OR) of 1.3 (95% confidence interval [CI]: 1.09, 1.35) for delivering preterm (<37 weeks) compared with women who screened negative [4]. Infants born to mothers with prenatal depression have been reported to have a lower mean birth weight than those born to non-depressed mothers, with a relative risk of 1.9 (95% CI: 1.3, 2.9) for low birth weight (≤2500 g) [6]. Furthermore, depressive and anxiety symptoms in early pregnancy have been associated with risk for subsequent preeclampsia, with adjusted ORs of 2.5 (95% CI: 1.1,5.4) for depression, 3.2 (95% CI: 1.4, 7.4) for anxiety, and 3.1 (95% CI: 1.4, 6.9) for women with either depression or anxiety [7]. Moreover, according to the perinatal programming hypotheses, depressive and anxious mood during pregnancy can affect the long-term behavioral and emotional development of children [8,9,10,11]. In a large prospective cohort study, children whose mothers had high levels of anxiety during late pregnancy demonstrated a higher likelihood of behavioral and emotional problems at 81 months of age (approximately 6.8 years), with ORs of 1.91 (95% CI: 1.26, 2.89) for girls and 2.16 (95% CI: 1.41, 3.30) for boys [9]. Prenatal mental disorders are also associated with a higher risk of postpartum mental disorders [12,13,14,15] and complications such as bonding failure [16,17]. Appropriate interventions and different approaches are required, in addition to early detection of these disorders. Among these, dietary modification has garnered increasing attention as a potentially modifiable factor.

Dietary habits affect not only perinatal complications but also mental health, including depression and anxiety during pregnancy and the postpartum period. Several studies have shown that nutrients and foods, such as vitamin B12, omega-3 polyunsaturated fatty acids (n-3 PUFAs), fish, seaweed, and yogurt, protect against perinatal mental disorders [18,19,20,21,22]. These nutrients and foods may exert beneficial effects through various molecular mechanisms, including antioxidant and anti-inflammatory actions, as well as modulation of neurotransmitter synthesis and neurotrophic support, which may contribute to mood regulation and a reduced risk of depression and anxiety [23,24]. However, the intake of individual nutritional factors and foods is complex, and the bioavailability of different nutrients can be influenced by the presence or absence of many dietary factors [25,26]. Recent studies have focused on dietary patterns to address these issues.

The Mediterranean diet (MD) is one of the most well-recognized healthy dietary patterns worldwide. MD is a traditional diet in the Mediterranean coastal areas (e.g., Greece, Italy, and Spain), characterized by high consumption of vegetables, fruits, legumes, fish, olive oil (particularly extra virgin olive oil), and cereals, moderate intake of milk and dairy products, and low consumption of meat and meat products [27,28]. MD was inscribed on the UNESCO Representative List of the Intangible Cultural Heritage of Humanity in 2013 and remains a cornerstone of dietary recommendations for both healthy individuals and patients with various health conditions [29]. The health benefits of MD have been extensively reported, including reduced risks of cardiovascular and total mortality [30], cancer [31], type 2 diabetes [32], and metabolic syndromes [33]. Recent studies have suggested that MD may be beneficial in the prevention of perinatal mental disorders [34,35,36,37,38]. Nevertheless, because of the variety of factors involved in perinatal mental disorders, the sample sizes were limited to exclude confounding factors in previous studies. Although several studies have been conducted in the Mediterranean region [34,35,38,39], reports from non-Mediterranean regions remain limited, and no such studies have been conducted in Japan.

Various MD scoring methods have been developed to assess adherence to MD. These methods can be broadly classified as relative and absolute scoring systems. Representative relative scoring systems were based on the median intake of dietary components within the study population. Examples include the Mediterranean Diet Score (MDS) [28], alternate Mediterranean Diet Score (aMED) [40], relative Mediterranean Diet (rMED) [41], and Mediterranean Diet Score in Pregnancy (PMDS) [42]. In contrast, absolute scoring systems are based on the frequency of consumption of specific food groups, with the MedDietScore [43] being one of the most frequently used. Among these, the MDS and aMED are the most widely employed. PMDS is an adapted form of MDS specifically for pregnancy. Currently, there is no consensus regarding the superiority of one scoring system over another, as the choice often depends on the study objectives, population characteristics, and geographic region under investigation.

In this study, we aimed to investigate the association between adherence to MD and psychological distress during pregnancy using data from the Japan Environment and Children’s Study (JECS), a large nationwide birth cohort study of Japanese pregnancies. In addition, we examined the population attributable fraction (PAF) of low MD scores for psychological distress to provide a quantitative target for the effective implementation of public health policies. Understanding the association between MD scores and psychological distress during pregnancy may help to develop novel approaches for the management and prevention of depressive and anxiety symptoms during pregnancy.

## 2. Materials and Methods

### 2.1. Study Design

We used data from the JECS to explore the association and PAF of adherence to MD during pregnancy with psychological distress during the second/third trimesters (MT-2).

The JECS is an ongoing, prospective nationwide birth cohort study in Japan that aims to identify environmental factors affecting children’s health and development. Approximately 100,000 pregnant women (and fathers, if accessible) were recruited from 15 Regional Centres (Hokkaido, Miyagi, Fukushima, Chiba, Kanagawa, Koshin, Toyama, Aichi, Kyoto, Osaka, Hyogo, Tottori, Kochi, Fukuoka, and South Kyusyu/Okinawa) from January 2011 to March 2014. The JECS protocol was reviewed and approved by the Ministry of the Environment’s Institutional Review Board on Epidemiological Studies and the Ethics Committees of all participating institutions.

In the JECS, questionnaires were administered to pregnant women in their first trimester (MT-1) and MT-2. The median and interquartile range (IQR) of gestational ages for the questionnaire responses were 15.4 (12.4, 19.4) weeks (MT-1) and 27.7 (25.4, 30.4) weeks (MT-2), respectively. We analyzed the data using the latest dataset released by the JECS Programme Office in March 2025. The baseline characteristics and study protocol of the participants have been described in detail previously [44,45].

### 2.2. Exposure (MD Score)

We used three MD scores—namely, the MDS [28], rMED [41], and PMDS [42]. These scores were used in a previous study on MD scores using the JECS data [46].

In the MDS, a value of 0 or 1 was assigned to each of the nine dietary components using the sex-specific median as the cutoff. For beneficial components (vegetables, fruits, nuts, fish, cereals, legumes, and the ratio of monounsaturated lipids to saturated lipids), each intake below the median was assigned a value of 0, and a value of 1 was assigned at or above the median. For detrimental components (dairy products and meat), each intake below the median was assigned a value of 1, and those at or above the median were assigned a value of 0. For alcohol intake, a value of 1 was assigned to women who consumed 5–25 g/day (ethanol equivalent). The total MDS ranges from 0 to 9 points [28]. In accordance with previous studies, we divided the participants into high (≥5 points) and low (≤4 points) MDS groups [28,46].

The rMED is a variation in the MDS, based on the same nine components. Values of 0, 1, and 2 were assigned to the first, second, and third intake tertiles, respectively. Unlike the MDS, the fat component was evaluated based on olive oil intake. The final score ranged from 0 to 18 points. Details of the rMED have been previously described [41]. In accordance with previous studies, we divided the participants into high (≥11 points) and low (≤10 points) rMED groups [41,46].

The PMDS is a modified version of the MDS specifically designed for use during pregnancy. In contrast to the MDS, dairy products are categorized as beneficial components of this scoring system. Furthermore, alcohol and fat ratios were excluded. The final PMDS ranged from 0 to 7 points. Details of PMDS have been described previously [42]. In accordance with previous studies, participants were divided into high (≥4 points) and low (≤3 points) MDS groups [42,46].

Each reference value in the previous reports was obtained from a population living in the Mediterranean region [28,41,42]. The current study was conducted in a Japanese population, and it is unclear whether the reference values of the study population are appropriate for assessing adherence to the MD. Therefore, we selected reference values obtained from Mediterranean populations in previous studies [28,41,42].

Information on maternal dietary intake necessary for scoring was obtained using the Food Frequency Questionnaire (FFQ) at MT-1 and MT-2 [47]. The FFQ at MT-1 was designed to assess pre-pregnancy dietary habits, while the FFQ at MT-2 was designed to assess dietary habits during pregnancy. The estimated daily intake (g/day) of each food item was calculated by multiplying the frequency of intake (per day) by the portion size of each food intake (g) in the questionnaire. The estimated daily energy intake (kcal/day) for each food item was calculated by multiplying the energy content by the energy intake [46].

### 2.3. Outcome (Psychological Distress)

In this study, the primary outcome was “psychological distress,” defined based on the Kessler 6-item Psychological Distress Scale (K6) score.

The K6 is a widely used screening tool for mental disorders in the general population [48,49]. It has been reported to effectively detect major depression and mood dysthymia, according to the DSM-IV along with the K10 [49]. The Japanese version of the K6 was developed using the standard back-translation method [50]. This scale comprises six questions, each with five possible responses (0–4 points) for each question, with a total score ranging from 0 to 24. In accordance with previous reports, we classified women with K6 scores ≥13 as experiencing psychological distress [51,52,53,54]. The Japanese version has demonstrated high reliability and validity [50].

The K6 was collected from the MT-2 questionnaire.

### 2.4. Collection and Classification of Other Variables

Using the JECS dataset, we collected data on maternal age, multiple pregnancy, regions where Regional Centres exist, pre-pregnancy height and body weight, alcohol intake status, smoking history of the couple, diabetes mellitus (DM), marital status, and employment status using the MT-1 questionnaire. Physicians, midwives/nurses, and/or Research Co-ordinators transcribed information on parity, conception method, and gestational diabetes mellitus (GDM).

The highest level of education for couples and the annual household income were obtained using the MT-2 questionnaire.

Late-pregnancy weight (measured after 28 weeks of gestation), which is required for calculating gestational weight gain, was obtained from the maternity health records transcribed by healthcare professionals.

Classifications of the other variables are listed in Table 1.

### 2.5. Statistical Analysis

Statistical analyses were conducted using the Statistical Analysis System (SAS) software (version 9.4; SAS Institute, Cary, NC, USA) and R version 4.3.0 (R Foundation for Statistical Computing, Vienna, Austria). Continuous variables are presented as medians (IQR). Categorical variables are presented as counts (percentages).

Differences in the proportions of psychological distress between the high and low groups for each MD score were compared using Pearson’s chi-square test. Statistical significance was set at a two-sided *p*-value < 0.05.

A modified Poisson regression model was used to calculate the risk ratios (RRs) and their 95% CIs, which were determined to assess the association between MD scores during pregnancy and psychological distress at MT-2 [55]. Furthermore, we estimated PAF to quantify the proportion of psychological distress that could have been prevented in the absence of low MD scores during pregnancy. Using the formula described below, we calculated the PAF and 95% CI using the SAS nonlinear estimate (NLEST) macro [56].*Estimated excess psychological distress case *=* Pe(RRe *− 1*)/RRe**PAF *=* estimated excess psychological distress case/all psychological distress cases* × 100
where *Pe* represents the proportion of psychological distress cases within each category, and *RRe* denotes the RR of psychological distress associated with exposure to low MD scores during pregnancy [57,58].

For the analysis of RR in the association between low MD scores during pregnancy and psychological distress, the high MD score groups were used as references. In the analysis of the PAF for the risk of psychological distress, high MD score groups were set as the reference categories.

Two models were constructed to calculate RRs and PAFs. The first model was defined as the crude model. The covariates for the adjusted model were selected, as shown in the directed acyclic graph in Figure 1. The model was adjusted for maternal age at MT-1 [59,60,61], pre-pregnancy body mass index (BMI) [38,62,63], parity [61,64,65], multiple pregnancy [66], conception method [67], DM or GDM [68,69], alcohol intake status [70,71], smoking history of a couple [70,71], the highest level of education for a couple [61], annual household income [70,72], marital status [73], employment status [70], regions where Regional Centres exist [74], and MDS at MT-1 [75]. All the covariates were included as categorical variables. Although mental status (MT-1 K6 score) in early pregnancy is thought to influence mental status during mid-to-late pregnancy, the MT-1 K6 score was not adjusted for in the current study because it was greatly affected by hyperemesis, nausea, and vomiting during pregnancy.

Additionally, modified Poisson regression models were used to simultaneously compare the strengths of the associations between each MD score on MT-2 and psychological distress, including the other MD scores (PMDS vs. MDS, PMDS vs. rMED, and MDS vs. rMED).

Missing covariate values were completed using the single imputation method with k-nearest neighbors in the R simulation package [76]. A general linear model was used to assess multicollinearity among the covariates, and all variance inflation factors were <10.

## 3. Results

### 3.1. Participant Selection

A flowchart of the participant selection is shown in Figure 2. The JECS included 104,043 pregnancies. Only initial participation data were used, resulting in a study population of 97,392 pregnant women. However, participants who had an abortion or stillbirth, missing data or improbable data, and calorie intake outside the ±2SD range (216.0–3253.9 kcal/day) were excluded from the analysis. To target mentally healthy pregnant women, those with a history of mental ill-ness were also excluded. Thus, the study included 80,271 participants, accounting for 77.2% of all study participants.

### 3.2. Characteristics of the Study Participants

Maternal and paternal characteristics of the study participants are summarized in Table 1. The median (IQR) age and pre-pregnancy BMI were 31 (27, 34) years and 20.5 (19.1, 22.5) kg/m^2^, respectively. Most participants were married (95.3%) and had a college education or higher (maternal, 64.5%; paternal, 56.2%). Approximately half of the participants had an annual income above the national average in Japan [77].

The median (IQR) MDS, rMED, and PMDSs were 4 (3, 4), 8 (6, 9), and 4 (3, 5) points, respectively. In the MDS analysis, 60,803 (75.7%) pregnant women were assigned to the low-scoring group, while 19,468 (24.3%) were assigned to the high-scoring group. In the rMED analysis, 73,786 (91.9%) pregnant women were included in the low rMED group and 6485 (8.1%) were in the high rMED group. In the PMDS analysis, 26,414 (32.9%) pregnant women were classified into the low group and 53,857 (67.1%) into the high group.

### 3.3. Main Results

#### 3.3.1. Association Between MD Scores and K6 Points

Table 2 shows the differences in the total K6 scores at MT-2 between the low- and high-MDS, rMED, and PMDS groups. At MT-2, total K6 scores of 4 or less, 5–9, and 10–12 at MT-2 were observed in 58,569 (73.0%), 14,630 (18.2%), and 5001 (6.2%) women, respectively. Overall, 2071 (2.6%) pregnant women had K6 scores ≥13 (psychological distress). Across all MD scoring systems, women with a high MD score were more likely to have a K6 score of ≤4, whereas those with a K6 score ≥5 were more likely to belong to the low MD score group.

A trend toward a higher proportion of psychological distress was observed in the low MDS group compared with the high MDS group (*p* = 0.08). The number and proportion of participants with psychological distress in the low- and high-MDS groups were 1603 (2.6%) and 468 (2.4%), respectively (Table 2). No significant difference in the proportion of psychological distress was observed between the high and low rMED groups (*p* = 0.24). The number and proportion of psychological distress cases in the low and high rMED groups were 1918 (2.6%) and 153 (2.4%), respectively (Table 2). In contrast, the high PMDS group had a significantly lower proportion of psychological distress than the low PMDS group (*p* < 0.0001). The number and proportion of participants with psychological distress were 847 (3.2%) in the low group and 1224 (2.3%) in the high group (Table 2).

#### 3.3.2. Association Between MD Scores and Psychological Distress

The association between MD scores during pregnancy and psychological distress at MT-2 is illustrated in Figure 3a. The crude model revealed that the low MDS group did not show a statistically significant increase in the risk of psychological distress (crude RR [cRR]: 1.10; 95% CI: 0.99, 1.21). In the adjusted model, the low MDS group did not show a statistically significant increase in the risk of psychological distress (adjusted RR [aRR]: 1.03; 95% CI: 0.92, 1.14). Regarding rMED, the crude model indicated that the low rMED group did not show a statistically significant increase in the risk of psychological distress (cRR: 1.10; 95% CI: 0.94, 1.30). In the adjusted model, the low rMED group did not show a statistically significant increase in the risk of psychological distress (aRR: 0.97; 95% CI: 0.82, 1.14). Regarding PMDS, the crude model revealed that the low PMDS group exhibited a statistically significant increase in the risk of psychological distress (cRR: 1.41; 95% CI: 1.29, 1.54). After adjustment for covariates, the low PMDS group showed a statistically significant increase in the risk of psychological distress (aRR: 1.17; 95% CI: 1.07, 1.28).

The PAFs for the risk of psychological distress due to low MD scores are presented in Figure 3b. The crude PAF (cPAF) from the low MDS group was 2.29% (95% CI: −0.29, 4.87), and the adjusted PAF (aPAF) was 0.66% (95% CI: −1.94, 3.29). Given the cPAF from low rMED, the cPAF and aPAF were 0.82% (95% CI: −0.61, 2.24) and −0.26% (95% CI: −1.56, 1.04), respectively. Owing to the low PMDS, the cPAF and aPAF were 21.61% (95% CI: 16.59, 26.64) and 10.43% (95% CI: 4.81, 16.06), respectively.

#### 3.3.3. Comparison of Each MD Score in Psychological Distress

A comparison of the associations between each MD score during pregnancy and psychological distress at MT-2 in the adjusted models is shown in Figure 4.

When both PMDS and MDS were included in the models simultaneously, the adjusted RRs were 1.20 (95% CI: 1.09, 1.32) for PMDS and 0.94 (95% CI: 0.84, 1.06) for MDS. When both PMDS and rMED were included in the models simultaneously, the adjusted RRs were 1.18 (95% CI: 1.08, 1.29) for PMDS and 0.93 (95% CI: 0.79, 1.10) for rMED. Neither MDS nor rMED was associated with psychological distress when both MDS and rMED were included in the adjusted model simultaneously (Figure 4a).

When both PMDS and MDS were included in the models simultaneously, the adjusted PAFs were 11.73% (95% CI: 5.58, 17.88) for PMDS and −1.42% (95% CI: −4.16, 1.32) for MDS. When both PMDS and rMED were included in the models simultaneously, the adjusted PAFs were 10.79% (95% CI: 5.12, 16.46) for PMDS and −0.59% (95% CI: −1.85, 0.67) for rMED. The aPAFs for MDS and rMED were not statistically significant when both MDS and rMED were included in the adjusted models simultaneously (Figure 4b).

## 4. Discussion

To the best of our knowledge, this is the first study to examine the association between MD scores and psychological distress in a large Japanese cohort. Compared with the high PMDS group, the low PMDS group had a significantly higher risk of psychological distress. After adjusting for potential confounding factors, including maternal and paternal characteristics, socioeconomic factors, and geographic location, high adherence to the MD based on the PMDS may reduce psychological distress by 10.4%. However, no statistically significant differences in psychological distress were observed between the high- and low-scoring MDS and rMED groups.

Studies on the efficacy and safety of medications for perinatal mental disorders are limited. While no increased risk of congenital malformations has been reported for selective serotonin reuptake inhibitors, which are often prescribed as antidepressants during pregnancy [78,79], an increased risk of poor neonatal adaptation syndrome and persistent pulmonary hypertension in the newborn has also been reported [80,81,82,83]. Although antidepressant use is recommended for pregnant women depending on the symptom severity, dietary approaches could provide drug-free alternatives for managing perinatal mental health concerns. Adherence to MD based on the PMDS may help reduce psychological distress that requires drug therapy.

Several recent studies have reported the effects of MD on mental disorders during pregnancy. A previous randomized controlled trial with 1221 pregnant women showed that MD intervention was associated with a substantial reduction in mental anxiety and stress and improvements in sleep quality throughout gestation [34]. Similarly, in a longitudinal study of 152 pregnant women in Spain, higher adherence to MD was associated with better mental health during pregnancy [35]. To the best of our knowledge, only three studies have used MD scoring. A cross-sectional study that included 3941 postpartum women in Greece revealed that postpartum depression (evaluated using the EPDS) was markedly associated with lower levels of adherence to MD (assessed using the MedDietScore) [38]. Likewise, a cross-sectional survey conducted among 5314 pregnant women in Greece showed that reduced adherence to MD (assessed using the MedDietScore) was associated with perinatal depression [36]. Outside the Mediterranean region, in the United States, higher adherence to MD, as assessed by the aMED, was associated with lower depressive symptoms (evaluated by the PHQ-9) in pregnant women (N = 540) [37]. To the best of our knowledge, this study employed the largest cohort dataset compared with any study that used the MD scores. Although the scoring systems used differed, our PMDS findings were consistent with those of previous studies. In contrast, in MDS and rMED, adherence to MD did not result in a statistically significant risk reduction in psychological distress.

The main differences between the PMDS and MDS or rMED are the inclusion of alcohol-related items and the classification of dairy products. Some MD scores, such as the MDS and rMED, were originally developed for non-pregnant adult populations. Although some studies have modified these scores for pregnant women, for example, by excluding alcohol consumption and adjusting cutoff values for food group intakes (e.g., Oddo et al., 2023), it should be noted that such adaptations may still fail to fully capture pregnancy-specific physiological and nutritional changes, such as increased energy and micronutrient requirements [37]. Chatzi et al. proposed the PMDS [42], which excludes alcohol consumption and considers dairy products to be protective rather than detrimental, given that alcohol intake is prohibited during pregnancy and calcium requirements increase during this period. In this study, only 244 participants (0.3%) in the MDS and rMED groups reported alcohol intake (Appendix A). Additionally, many participants had scores near the cutoff between the high- and low-scoring groups (Appendix A). These findings suggest that applying the MDS and rMED to pregnant women may have resulted in a large proportion of participants being classified into low-scoring groups, potentially influencing the study results.

Previous studies have suggested that higher calcium intake is related to a lower prevalence of depressive symptoms [84,85,86]. In addition to calcium, short-chain fatty acids in yogurt and dairy products promote improvement in the intestinal microflora, and it has been reported that improvement in the intestinal environment may have an antidepressant effect [87,88]. Reports on the effects of dairy products on mental illness are limited and cannot be definitive. However, dairy products that were beneficial for PMDS may have contributed to the reduced risk of psychological distress in pregnant women with increased calcium requirements. Nevertheless, the scoring methods used during pregnancy warrant further investigation.

The mechanisms through which adherence to MD affects psychiatric symptoms are not fully understood. MD includes antioxidants (e.g., vitamins A, C, and E; polyphenols), *n*-3 PUFAs, B vitamins, monounsaturated fatty acids, vitamin D, and minerals including iodine, magnesium, zinc, selenium, potassium and iron [89]. Scapagnini et al. demonstrated the roles of various antioxidant nutrients, cofactors, and compounds in mood, cognition, and mental health. These compounds can prevent oxidative damage to cellular membranes or DNA in the central nervous system (CNS) and improve serotonin, dopamine, and glutathione levels, which appear to be modulated via markers of antioxidant activity [90]. The mechanisms of action of *n*-3 PUFAs in alleviating depressive symptoms are thought to involve anti-inflammatory effects, regulation of brain-derived neurotrophic factors, and improvement in cerebral blood flow [91]. The synergistic combination of these nutrients, which is positively associated with mental health among participants adhering to MD, may have provided greater benefits for mental health.

We analyzed the reference values obtained from the Mediterranean population. Consistent with a previous JECS analysis [46], we also performed an analysis using the median values of the current participants in the JECS; however, we did not observe a positive effect of high adherence to MD on the incidence of psychological distress in pregnant women. In the non-Mediterranean region, the intake of each component differs from that in the Mediterranean region, and the median or tertile values in these groups may not accurately reflect adherence to MD. Consequently, there may not have been a significant difference in the incidence of psychological distress between participants with high and low scores. Accordingly, selecting absolute measures based on the Mediterranean population rather than relative measures may be preferable for calculating MD compliance outside the Mediterranean region. Absolute scores, such as the MedDietScore, are based on the frequency of consumption of dietary components, while absolute scoring systems are rare [92]. Future studies should consider conducting quantitative absolute evaluations.

One of the strengths of our study is that the evaluation was conducted using PAFs, which led to the effective implementation of public health policies. Furthermore, this study involved in a large sample size, and participants were recruited from diverse areas of Japan. Therefore, the representativeness of the Japanese population was considered to be adequate. Various confounding factors, including paternal factors, were considered in the analyses. The MT-1 data included pre-pregnancy dietary habits, which allowed us to include pre-pregnancy dietary habits as confounding factors.

The limitations of this study also need to be addressed. First, because this cohort study was not originally designed for this specific purpose, the MD scoring systems used in previous reports could not be evaluated [37,38]. However, there is no definitive evidence regarding the most appropriate scoring system, and the choice of scoring also depends on the study population and the outcomes under investigation. We evaluated the three scoring schemes previously used in the JECS analysis [46]. The PMDS was specifically developed for pregnant women to better reflect pregnancy-related physiological and dietary characteristics. Second, baseline dietary patterns and psychological distress before MT-1 administration may have influenced the observed associations. Although information on pre-pregnancy dietary habits was available in the JECS dataset, data on pre-pregnancy psychological status (e.g., K6 score) were not collected. To address the potential confounding factors, we adjusted for the MT-1 MD score in the analysis. Furthermore, participants with histories of mental disorders were excluded to minimize bias. Nevertheless, the lack of pre-pregnancy psychological data remains a limitation. Third, all questionnaires were self-administered and may therefore have been subject to reporting bias due to the subjectivity of the respondents. Finally, few studies have used PMDS. To the best of our knowledge, only two studies have been conducted using a PMDS. Nakano et al. reported a relationship between MD scores (MDS, PMDS, and rMED) during pregnancy and the incidence of type 1 allergies in the offspring [46]. Chatzi et al. demonstrated the effect of maternal and child adherence to MD on childhood asthma and atopy [42]. In both studies, the outcome was allergic disease in children, and no study reported the use of maternal mental disorders as an outcome. Nevertheless, for the reasons outlined above, the PMDS may be preferable for examining the relationship between MD adherence and maternal mental health outcomes. Further studies are required to evaluate the association between PMDS and mental disorders. In addition, future analyses using the JECS dataset should explore how maternal adherence to the MD and psychological distress during pregnancy influence neonatal outcomes such as birth weight and gestational age.

## 5. Conclusions

Our study revealed that greater adherence to the MD, as assessed using the PMDS, was associated with lower levels of psychological distress among pregnant women. Although further studies are needed, improved dietary patterns may reduce the risk of perinatal mental disorders and could be a powerful strategy for promoting a healthy perinatal environment.

## Figures and Tables

**Figure 1 nutrients-17-03697-f001:**
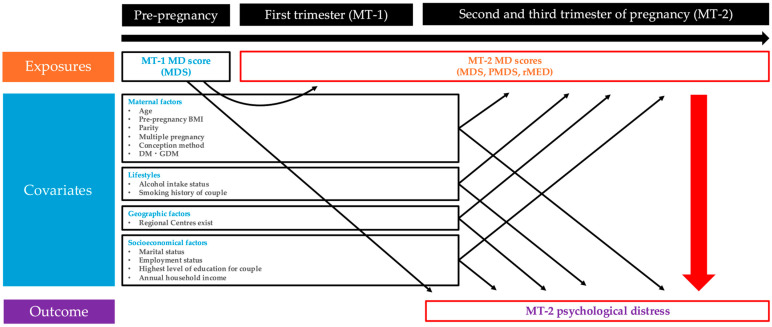
Directed acyclic graph of the association between MD scores during pregnancy with psychological distress at MT-2. BMI: body mass index, DM: diabetes mellitus, GDM: gestational diabetes mellitus, MD: Mediterranean diet, MDS: Mediterranean Diet Score, PMDS: Mediterranean Diet Score in Pregnancy, and rMED: relative Mediterranean Diet.

**Figure 2 nutrients-17-03697-f002:**
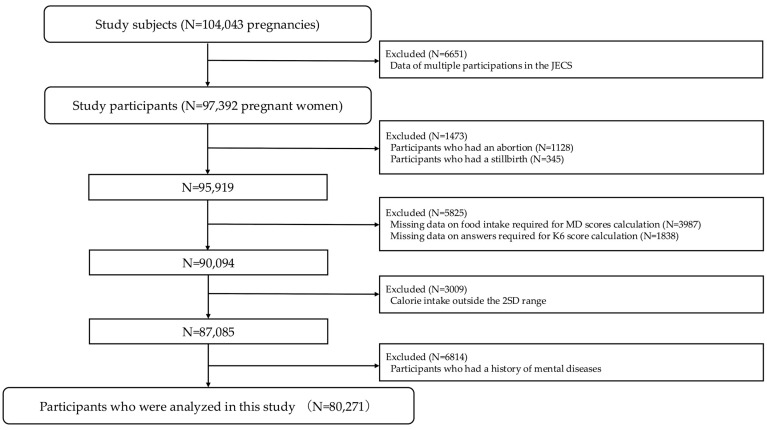
Flow chart of this study. JECS: Japan Environment and Children’s Study, K6: Kessler 6-item Psychological Distress Scale, MD: Mediterranean diet, and SD: standard deviation.

**Figure 3 nutrients-17-03697-f003:**
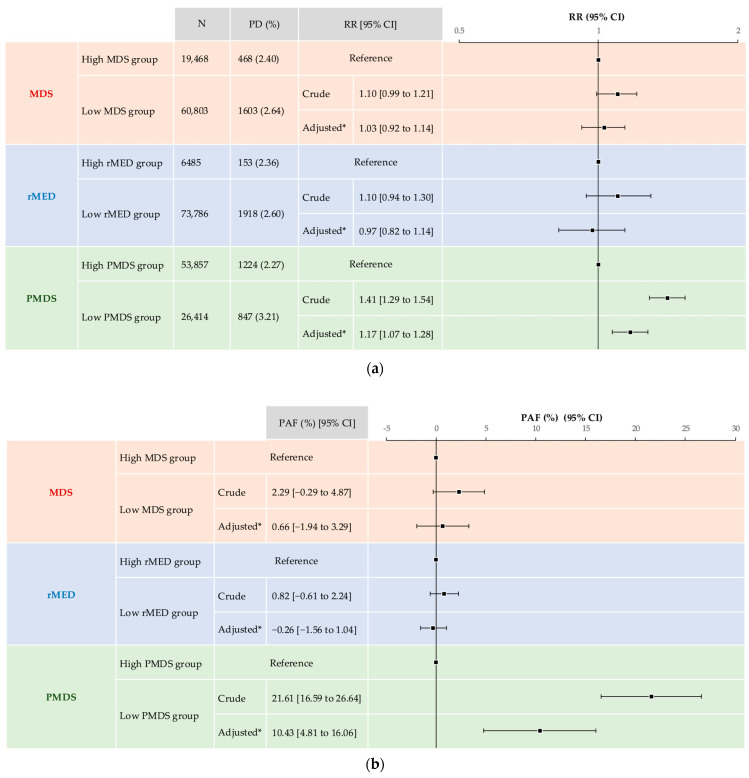
Association between MD scores during pregnancy with psychological distress at MT-2. (**a**) RRs of psychological distress at MT-2 due to low MD scores during pregnancy; (**b**) PAFs of psychological distress at MT-2 from low MD scores during pregnancy. * Adjusted models for maternal age at MT-1, pre-pregnancy BMI, parity, multiple pregnancy, conception method, DM/GDM, alcohol intake status, smoking history of couple, marital status, employment status, highest level of education for couple, annual household income, regions where Regional Centres exist, and MT-1 MDS. BMI: body mass index, CI: confidence interval, DM: diabetes mellitus, GDM: gestational diabetes mellitus, MD: Mediterranean diet, MDS: Mediterranean Diet Score, PAF: population attributable fraction, PD: psychological distress, PMDS: Mediterranean Diet Score in Pregnancy, rMED: relative Mediterranean Diet, and RR: risk ratio.

**Figure 4 nutrients-17-03697-f004:**
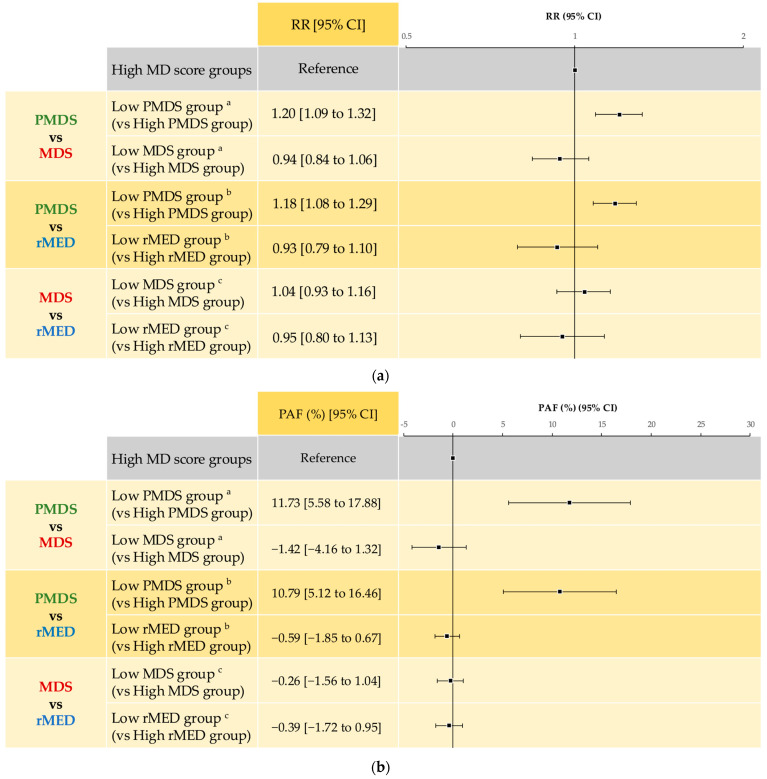
Comparison of the associations of each MD score during pregnancy with psychological distress at MT-2. (**a**) RRs of psychological distress at MT-2 due to low MD scores during pregnancy; (**b**) PAFs of psychological distress at MT-2 from low MD scores during pregnancy. ^a^ Both PMDS and MDS were included in the model simultaneously. Adjusted for maternal age at MT-1, pre-pregnancy BMI, parity, multiple pregnancy, conception method, DM/GDM, alcohol intake status, smoking history of the couple, marital status, employment status, highest level of education for couple, annual household income, regions where Regional Centres exist, and MT-1 MDS. ^b^ Both PMDS and rMED were included in the model simultaneously. Adjusted for maternal age at MT-1, pre-pregnancy BMI, parity, multiple pregnancy, conception method, DM/GDM, alcohol intake status, smoking history of the couple, marital status, employment status, highest level of education for couple, annual household income, regions where Regional Centres exist, and MT-1 MDS. ^c^ Both MDS and rMED were included in the model simultaneously. Adjusted for maternal age at MT-1, pre-pregnancy BMI, parity, multiple pregnancy, conception method, DM/GDM, alcohol intake status, smoking history of the couple, marital status, employment status, highest level of education for the couple, annual household income, regions where Regional Centres exist, and MT-1 MDS. BMI: body mass index, CI: confidence interval, DM: diabetes mellitus, GDM: gestational diabetes mellitus, MD: Mediterranean diet, MDS: Mediterranean Diet Score, PAF: population attributable fraction, PMDS: Mediterranean Diet Score in Pregnancy, rMED: relative Mediterranean Diet, and RR: risk ratio.

**Table 1 nutrients-17-03697-t001:** Baseline characteristics between high and low MD scoring groups (MDS, rMED, and PMDS). Continuous variables are expressed as the medians (IQR). Categorical variables are expressed as numbers (percentages). Low MDS = 0–4 points; High MDS = 5–9 points; Low rMED = 0–10 points; High rMED = 11–18 points; Low PMDS = 0–3 points; High PMDS = 4–7 points. GWG was calculated as the difference between late-pregnancy weight (measured after 28 weeks of gestation) and pre-pregnancy weight.

Variables	All Participants N = 80,271	MDS		rMED		PMDS	
		Low MDS N = 60,803	High MDS N = 19,468	Low rMED N = 73,786	High rMED N = 6485	Low PMDS N = 26,414	High PMDS N = 53,857
**Age at MT-1, y**	31 (27, 34)	31 (27, 34)	31 (28, 35)	31 (27, 34)	32 (29, 35)	30 (26, 33)	31 (28, 35)
**Age at MT-1, N (%)**							
<20 years	905 (1.1)	730 (1.2)	175 (0.9)	865 (1.2)	40 (0.6)	504 (1.9)	401 (0.7)
20–34.9 years	59,906 (74.6)	45,853 (75.4)	14,053 (72.2)	55,491 (75.2)	4415 (68.1)	20,877 (79.0)	39,029 (72.5)
≥35 years	19,446 (24.2)	14,209 (23.4)	5237 (26.9)	17,416 (23.6)	2030 (31.3)	5029 (19.0)	14,417 (26.8)
Missing	14 (0.02)	11 (0.02)	3 (0.02)	14 (0.02)	0 (0.0)	4 (0.02)	10 (0.02)
**Pre-pregnancy BMI, kg/m^2^**	20.5 (19.1, 22.5)	20.5 (19.1, 22.5)	20.5 (19.1, 22.5)	20.5 (19.1, 22.5)	20.5 (19.1, 22.6)	20.5 (19.1, 22.6)	20.5 (19.1, 22.4)
**Pre-pregnancy BMI, N (%)**							
Underweight (<18.5 kg/m^2^)	12,919 (16.1)	9699 (16.0)	3220 (16.5)	11,875 (16.1)	1044 (16.1)	4427 (16.8)	8492 (15.8)
Normal range (18.5–24.9 kg/m^2^)	58,972 (73.5)	44,807 (73.7)	14,165 (72.8)	54,264 (73.5)	4708 (72.6)	18,921 (71.6)	40,051 (74.4)
Overweight (≥25.0 kg/m^2^)	8347 (10.4)	6270 (10.3)	2077 (10.7)	7616 (10.3)	731 (11.3)	3053 (11.6)	5294 (9.8)
Missing	33 (0.04)	27 (0.04)	6 (0.03)	31 (0.04)	2 (0.03)	13 (0.05)	20 (0.04)
**GWG, kg**	8.2 (6.0, 10.4)	8.2 (6.1, 10.5)	8.1 (6.0, 10.3)	8.2 (6.1, 10.5)	7.9 (5.8, 10.0)	8.3 (6.1, 10.7)	8.1 (6.0, 10.3)
**Parity, N (%)**							
Multiparity	44,851 (55.9)	32,918 (54.1)	11,933 (61.3)	40,956 (55.5)	3895 (60.1)	14,020 (53.1)	30,831 (57.2)
Nulliparity	33,427 (41.6)	26,320 (43.3)	7107 (36.5)	30,985 (42.0)	2442 (37.7)	11,720 (44.4)	21,707 (40.3)
Missing	1993 (2.5)	1565 (2.6)	428 (2.2)	1845 (2.5)	148 (2.3)	674 (2.6)	1319 (2.4)
**Multiple pregnancy, N (%)**							
Singleton	79,480 (99.0)	60,202 (99.0)	19,278 (99.0)	73,075 (99.0)	6405 (98.8)	26,176 (99.1)	53,304 (99.0)
Multiple	791 (1.0)	601 (1.0)	190 (1.0)	711 (1.0)	80 (1.2)	238 (0.9)	553 (1.0)
**Conception method, N (%)**							
Spontaneous pregnancy	74,307 (92.6)	56,335 (92.7)	17,972 (92.3)	68,401 (92.7)	5906 (91.1)	24,950 (94.5)	49,357 (91.6)
Non-ART	3058 (3.8)	2336 (3.8)	722 (3.7)	2784 (3.8)	274 (4.2)	794 (3.0)	2264 (4.2)
ART	2549 (3.2)	1860 (3.1)	689 (3.5)	2267 (3.1)	282 (4.3)	560 (2.1)	1989 (3.7)
Missing	357 (0.4)	272 (0.4)	85 (0.4)	334 (0.5)	23 (0.4)	110 (0.4)	247 (0.5)
**DM/GDM, N (%)**	2350 (2.9)	1769 (2.9)	581 (3.0)	2141 (2.9)	209 (3.2)	785 (3.0)	1565 (2.9)
Missing	377 (0.5)	290 (0.5)	87 (0.4)	348 (0.5)	29 (0.4)	122 (0.5)	255 (0.5)
**Alcohol intake status, N (%)**							
Never or Quit drinking before	71,856 (89.5)	54,611 (89.8)	17,245 (88.6)	66,108 (89.6)	5748 (88.6)	23,865 (90.3)	47,991 (89.1)
Continue drinking	8090 (10.1)	5931 (9.8)	2159 (11.1)	7370 (10.0)	720 (11.1)	2406 (9.1)	5684 (10.6)
Missing	325 (0.4)	261 (0.4)	64 (0.3)	308 (0.4)	17 (0.3)	143 (0.5)	182 (0.3)
**Smoking history, N (%)**							
**Maternal**							
Not currently smoking	76,257 (95.0)	57,711 (94.9)	18,546 (95.3)	70,000 (94.9)	6257 (96.5)	24,516 (92.8)	51,741 (96.1)
Currently smoking	3462 (4.3)	2660 (4.4)	802 (4.1)	3270 (4.4)	192 (3.0)	1674 (6.3)	1788 (3.3)
Missing	552 (0.7)	432 (0.7)	120 (0.6)	516 (0.7)	36 (0.6)	224 (0.8)	328 (0.6)
**Paternal**							
Not currently smoking	42,148 (52.5)	31,811 (52.3)	10,337 (53.1)	38,350 (52.0)	3798 (58.6)	12,287 (46.5)	29,861 (55.4)
Currently smoking	36,603 (45.6)	27,818 (45.8)	8785 (45.1)	34,023 (46.1)	2580 (39.8)	13,488 (51.1)	23,115 (42.9)
Missing	1520 (1.9)	1174 (1.9)	346 (1.8)	1413 (1.9)	107 (1.6)	639 (2.4)	881 (1.6)
**Marital status, N (%)**							
Married	76,485 (95.3)	57,818 (95.1)	18,667 (95.9)	70,213 (95.2)	6272 (96.7)	24,620 (93.2)	51,865 (96.3)
Unmarried	2846 (3.5)	2227 (3.7)	619 (3.2)	2682 (3.6)	164 (2.5)	1334 (5.1)	1512 (2.8)
Divorced or widowed	611 (0.8)	489 (0.8)	122 (0.6)	575 (0.8)	36 (0.6)	308 (1.2)	303 (0.6)
Missing	329 (0.4)	269 (0.4)	60 (0.3)	316 (0.4)	13 (0.2)	152 (0.6)	177 (0.3)
**Employment status, N (%)**							
Not working	28,108 (35.0)	21,122 (34.7)	6986 (35.9)	25,806 (35.0)	2302 (35.5)	8740 (33.1)	19,368 (36.0)
Working	49,637 (61.8)	37,764 (62.1)	11,873 (61.0)	45,673 (61.9)	3964 (61.1)	16,830 (63.7)	32,807 (60.9)
Missing	2526 (3.1)	1917 (3.2)	609 (3.1)	2307 (3.1)	219 (3.4)	844 (3.2)	1682 (3.1)
**Highest education level, N (%)**							
**Maternal**							
Junior high school	3446 (4.3)	2669 (4.4)	777 (4.0)	3256 (4.4)	190 (2.9)	1682 (6.4)	1764 (3.3)
High school	24,799 (30.9)	18,902 (31.1)	5897 (30.3)	23,140 (31.4)	1659 (25.6)	9664 (36.6)	15,135 (28.1)
College	51,759 (64.5)	39,026 (64.2)	12,733 (65.4)	47,152 (63.9)	4607 (71.0)	14,964 (56.7)	36,795 (68.3)
Missing	267 (0.3)	206 (0.3)	61 (0.3)	238 (0.3)	29 (0.4)	104 (0.4)	163 (0.3)
** Paternal**							
Junior high school	5504 (6.9)	4263 (7.0)	1241 (6.4)	5197 (7.0)	307 (4.7)	2528 (9.6)	2976 (5.5)
High school	28,931 (36.0)	21,947 (36.1)	6984 (35.9)	26,735 (36.2)	2196 (33.9)	10,501 (39.8)	18,430 (34.2)
College	45,096 (56.2)	34,002 (55.9)	11,094 (57.0)	41,160 (55.8)	3936 (60.7)	13,052 (49.4)	32,044 (59.5)
Missing	740 (0.9)	591 (1.0)	149 (0.8)	694 (0.9)	46 (0.7)	333 (1.3)	407 (0.8)
**Annual household** **income, N (%)**							
<200 × 10^4^ JPY	3984 (5.0)	3115 (5.1)	869 (4.5)	3732 (5.1)	252 (3.9)	1873 (7.1)	2111 (3.9)
200–399 × 10^4^ JPY	25,388 (31.6)	19,420 (31.9)	5968 (30.7)	23,600 (32.0)	1788 (27.6)	9412 (35.6)	15,976 (29.7)
400–599 × 10^4^ JPY	24,913 (31.0)	18,701 (30.8)	6212 (31.9)	22,868 (31.0)	2045 (31.5)	7588 (28.7)	17,325 (32.2)
≥600 × 10^4^ JPY	20,521 (25.6)	15,416 (25.4)	5105 (26.2)	18,564 (25.2)	1957 (30.2)	5385 (20.4)	15,136 (28.1)
Missing	5465 (6.8)	4151 (6.8)	1314 (6.7)	5022 (6.8)	443 (6.8)	2156 (8.2)	3309 (6.1)
**Regions where Regional Centers exist, N (%)**							
Hokkaido	6531 (8.1)	5116 (8.4)	1415 (7.3)	6155 (8.3)	376 (5.8)	2123 (8.0)	4408 (8.2)
Tohoku	17,631 (22.0)	13,056 (21.5)	4575 (23.5)	15,886 (21.5)	1745 (26.9)	5612 (21.2)	12,019 (22.3)
Kanto	9592 (11.9)	7434 (12.2)	2158 (11.1)	8868 (12.0)	724 (11.2)	2887 (10.9)	6705 (12.4)
Chubu	14,644 (18.2)	10,800 (17.8)	3844 (19.7)	13,272 (18.0)	1372 (21.2)	4335 (16.4)	10,309 (19.1)
Kinki	13,286 (16.6)	10,399 (17.1)	2887 (14.8)	12,489 (16.9)	797 (12.3)	4391 (16.6)	8895 (16.5)
Chugoku	2324 (2.9)	1731 (2.8)	593 (3.0)	2093 (2.8)	231 (3.6)	694 (2.6)	1630 (3.0)
Shikoku	5586 (7.0)	4158 (6.8)	1428 (7.3)	5139 (7.0)	447 (6.9)	2139 (8.1)	3447 (6.4)
Kyusyu-Okinawa	10,677 (13.3)	8109 (13.3)	2568 (13.2)	9884 (13.4)	793 (12.2)	4233 (16.0)	6444 (12.0)
**MDS at MT-2, points**	4 (3, 4)						
**MDS at MT-2, N (%)**							
Low MDS	60,803 (75.7)	-	-	58,896 (79.8)	1907 (29.4)	26,085 (98.8)	34,718 (64.5)
High MDS	19,468 (24.3)	-	-	14,890 (20.2)	4578 (70.6)	329 (1.2)	19,139 (35.5)
**rMED at MT-2, points**	8 (6, 9)						
**rMED at MT-2, N (%)**							
Low rMED	73,786 (91.9)	58,896 (96.9)	14,890 (76.5)	-	-	25,815 (97.7)	47,971 (89.1)
High rMED	6485 (8.1)	1907 (3.1)	4578 (23.5)	-	-	599 (2.3)	5886 (10.9)
**PMDS at MT-2, points**	4 (3, 5)						
**PMDS at MT-2, N (%)**							
Low PMDS	26,414 (32.9)	26,085 (42.9)	329 (1.7)	25,815 (35.0)	599 (9.2)	-	-
High PMDS	53,857 (67.1)	34,718 (57.1)	19,139 (98.3)	47,971 (65.0)	5886 (90.8)	-	-

ART: assisted reproductive technology, BMI: body mass index, DM: diabetes mellitus, GDM: gestational diabetes mellitus, GWG: gestational weight gain, IQR: interquartile range, MDS: Mediterranean Diet Score, PMDS: Mediterranean Diet Score in Pregnancy, and rMED: relative Mediterranean Diet.

**Table 2 nutrients-17-03697-t002:** Total K6 score at MT-2 between high and low MD scoring groups (MDS, rMED, and PMDS). Low MDS = 0–4 points; High MDS = 5–9 points; Low rMED = 0–10 points; High rMED = 11–18 points; Low PMDS = 0–3 points; High PMDS = 4–7 points.

Total K6 Score at MT-2	All Participants N = 80,271	Low MDS N = 60,803	High MDS N = 19,468	*p*	Low rMED N = 73,786	High rMED N = 6485	*p*	Low PMDS N = 26,414	High PMDS N = 53,857	*p*
≤4, N (%)	58,569 (73.0)	44,095 (72.5)	14,474 (74.3)		53,755 (72.9)	4814 (74.2)		18,622 (70.5)	39,947 (74.2)	
5–9, N (%)	14,630 (18.2)	11,205 (18.4)	3425 (17.6)		13,494 (18.3)	1136 (17.5)		5015 (19.0)	9615 (17.9)	
10–12, N (%)	5001 (6.2)	3900 (6.4)	1101 (5.7)		4619 (6.3)	382 (5.9)		1930 (7.3)	3071 (5.7)	
≥13, N (%) (Psychological Distress)	2071 (2.6)	1603 (2.6)	468 (2.4)	0.08	1918 (2.6)	153 (2.4)	0.24	847 (3.2)	1224 (2.3)	<0.0001

K6: Kessler psychological distress scale, MD: Mediterranean diet, MDS: Mediterranean Diet Score, PMDS: Mediterranean Diet Score in Pregnancy, and rMED: relative Mediterranean Diet.

## Data Availability

The data presented in this study are available on request from the corresponding author due to ethical restrictions and legal framework of Japan.

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
