# Peer review of "Association of Mediterranean Diet Scores with Psychological Distress in Pregnancy: The Japan Environment and Children’s Study"

_nutrients, 2025, doi:10.3390/nu17233697_

Round 1

Reviewer 1 Report

Comments and Suggestions for Authors

This is a well-organized research study with adequate novelty. However, some points should be addressed. 

  • In the Abstract, the authors should provide a short background of the topic of the study.
  • The Introduction section is quite short. The authors should firstly provide some relevant epidemiological data in the 1st paragraph.
  • In the 2nd paragraph of the Introduction section, the authors should report the most important molecular mechanisms through which nutrients and foods may exert beneficial effects in human health (e.g. antioxidant and anti-inflammatory properties, etc.).
  • The different MD scores should be reported in the Introduction section after the 3rd paragraph.
  • The Gestational Weight Gain (GWG) could be a significant factor for this study. Are there any data for GWG which may be assessed in comparison with MD adherance scores?
  • The normality distribution test should be reported in the section 2.5.
  • In section 3.1, the % final response rate should be reported.
  • Please revise the beggining of the sentence in line 249 (....points, respectively???).
  • English language editing is recommended.

Comments on the Quality of English Language

English language editing is recommended.

Author Response

Response to Reviewer 1 Comments

1. Summary

2. Questions for General Evaluation

Reviewer’s Evaluation

Response and Revisions

Does the introduction provide sufficient background and include all relevant references?

Can be improved

Are all the cited references relevant to the research?

Yes

Is the research design appropriate?

Can be improved

Are the methods adequately described?

Can be improved

Are the results clearly presented?

Yes

Are the conclusions supported by the results?

Yes

3. Point-by-point response to Comments and Suggestions for Authors

Comments 1: In the Abstract, the authors should provide a short background of the topic of the study.

Response 1: Thank you for your valuable comment. We agree with your suggestion and have added a brief background on the study topic at the beginning of the Abstract (page 1, lines 33–36) to provide context for the research.

Comments 2: The Introduction section is quite short. The authors should firstly provide some relevant epidemiological data in the 1st paragraph.

Response 2: Thank you for pointing this out. We agree with your suggestion. Accordingly, we have revised the first paragraph of the Introduction to include relevant epidemiological data and to better emphasize the public health importance of perinatal mental disorders.

Comments 3: In the 2nd paragraph of the Introduction section, the authors should report the most important molecular mechanisms through which nutrients and foods may exert beneficial effects in human health (e.g. antioxidant and anti-inflammatory properties, etc.).

Response 3: Thank you for this valuable comment. We agree with your suggestion and have revised the Introduction section to briefly describe the key molecular mechanisms through which nutrients and foods may exert beneficial effects on mental health, such as antioxidant, anti-inflammatory, and neuroprotective properties.

Comments 4: The different MD scores should be reported in the Introduction section after the 3rd paragraph.

Response 4: Thank you for pointing this out. We agree with your suggestion. Accordingly, we have added a description of the different Mediterranean diet (MD) scoring systems after the third paragraph in the Introduction section (page 3, lines 106–117).

Comments 5: The Gestational Weight Gain (GWG) could be a significant factor for this study. Are there any data for GWG which may be assessed in comparison with MD adherance scores?

Response 5: Thank you for this valuable comment. Previous research has reported an association between gestational weight gain (GWG) and postpartum depression (Papadopoulou et al., Nutrients, 2023), suggesting that GWG may also influence psychological distress during pregnancy. Data on GWG are available in the JECS dataset, and we have added this information to Table 1.

Based on our directed acyclic graph, GWG was considered an intervening variable in the causal pathway between the exposure (Mediterranean diet score at MT-2) and the outcome (psychological distress at MT-2). Therefore, adjusting for GWG could potentially attenuate the observed association by controlling for a mediator on the causal pathway.

Nevertheless, to further explore this possibility, we conducted additional analyses including GWG as a covariate. The results were essentially consistent with the main findings, suggesting that the association between adherence to the Mediterranean diet and psychological distress during pregnancy cannot be fully explained by GWG.

Regarding the calculation of GWG presented in Table 1, ideally, GWG would have been calculated as the difference between body weight at MT-2 and pre-pregnancy body weight. However, body weight at MT-2 was not measured in the present dataset. Therefore, GWG was calculated as the difference between body weight in late pregnancy (after 28 weeks of gestation) and pre-pregnancy body weight.

Comments 6: The normality distribution test should be reported in the section 2.5.

Response 6: Thank you for your comment. We agree that assessing data distribution is important. However, in a study with a large sample size exceeding 80,000 participants, normality tests such as the Kolmogorov–Smirnov test tend to detect even trivial deviations from normality that are not meaningful in practice. Therefore, instead of conducting formal normality tests, all continuous variables were described using medians and interquartile ranges (IQRs) in Table 1.

Comments 7: In section 3.1, the % final response rate should be reported.

Response 7: Thank you for pointing this out. We agree with this comment. We have added the final response rate percentage and modified the sentence (page 10, line 282–283).

Comments 8: Please revise the beginning of the sentence in line 249 (....points, respectively???).

Response 8: Thank you for bringing this to our attention. The interruption in the text was due to an error that occurred during the formatting and layout adjustment of the table and surrounding text. We sincerely apologize for this oversight. The missing portion has now been restored in line 294 to ensure the continuity and accuracy of the text.

Comments 9: English language editing is recommended.

Response 9: Thank you for pointing this out. The submitted manuscript was professionally proofread by a native English editor at Editage on July 4, 2025 (Job Code: MFNBR_3). We have also had the revised version reproofread to further improve clarity and accuracy (November 10, 2025; Job Code: MFNBR_3_2). We would like to respectfully ask the reviewers and editors to assess the current language quality. If any additional revisions are deemed necessary, we will be happy to make them promptly.

4. Response to Comments on the Quality of English Language

Point 1: English language editing is recommended.

Response 1: As stated above, the manuscript has been professionally proofread by a native English editor at Editage, and the revised version has also undergone additional proofreading to further ensure language quality.

5. Additional clarifications

None

Reviewer 2 Report

Comments and Suggestions for Authors

In this manuscript, Takahashi et. al investigate the association between adherence to the Mediterranean diet during pregnancy and psychological distress in a large Japanese birth cohort. The study leverages a robust sample size (n = 80,271) from the Japan Environment and Children’s Study, assessing adherence through three scoring systems (MDS, rMED, and PMDS). The use of modified Poisson regression and the calculation of population attributable fractions are appropriate and well described.

Several methodological and interpretative issues should be addressed before the manuscript can be considered for publication. Clarifications regarding diet scoring, baseline psychological status, data continuity, and interpretation of borderline statistical results are required. The manuscript would also benefit from improved referencing, formatting consistency, and careful revision for precision and presentation.

Major comments

Please clarify the definition of “low consumption of meat and meat products.”

As some of the diet scores (MDS and rMED) were originally designed for non-pregnant adults, their direct application to pregnant women may not fully capture physiological or dietary adaptations during pregnancy. Please justify or discuss the implications of this limitation.

The sentences “Many studies have reported the health benefits of MD, including reduced risks of cardiovascular and total mortality, cancer, type 2 diabetes, and metabolic syndrome,” and “Several studies have been conducted on the Mediterranean region” require appropriate references. Including recent meta-analyses or cohort studies would strengthen the introduction.

It would be valuable to know whether any pre-pregnancy dietary or psychological data were available. Could baseline psychological status before MT-1 have influenced the observed associations?

Were the same women followed at MT-2, and if so, was there any indication of pre-existing mental vulnerability?

Have the authors explored whether using only extreme quartiles (1st vs 4th) for each score would meaningfully change the results? This could highlight direct effects or enhance contrast between groups.

Are there available data linking maternal adherence to MD or psychological distress with neonatal outcomes (e.g., birth weight, gestational age, APGAR score)? Even a brief discussion of these potential associations would enhance the manuscript’s translational value.

The statement “No significant differences in the proportion of psychological distress were found between the high and low MDS groups (p = 0.08)” requires more cautious interpretation. A p-value of 0.08 may indicate a trend toward significance rather than the absence of an effect. Furthermore, the used statistical test aims at proving that two groups are different, thus, a lack of statistical significance does not mean that two groups are equal. Please revise the wording accordingly and avoid overstating null results.

Between lines 248 and 249, the text appears interrupted. Please restore or clarify the missing portion.

Minor comments

Move complete table legends above the respective tables for consistency with journal formatting standards.

Ensure alignment and text formatting are uniform throughout the manuscript, as several unformatted or misaligned passages are present.

Carefully proofread for minor typographical or spacing issues.

Author Response

Response to Reviewer 2 Comments

1. Summary

2. Questions for General Evaluation

Reviewer’s Evaluation

Response and Revisions

Does the introduction provide sufficient background and include all relevant references?

Can be improved

Are all the cited references relevant to the research?

Yes

Is the research design appropriate?

Yes

Are the methods adequately described?

Yes

Are the results clearly presented?

Can be improved

Are the conclusions supported by the results?

Can be improved

3. Point-by-point response to Comments and Suggestions for Authors

Comments 1: Please clarify the definition of “low consumption of meat and meat products.”

Response 1: Thank you for pointing this out. The definition of “low consumption of meat and meat products” follows the description provided by Trichopoulou et al. (Reference No.  27). In addition, a detailed study from the European EPIC cohorts (Linseisen et al., Public Health Nutrition, 2002) reported that meat intake in Mediterranean populations is generally lower than in other dietary patterns, such as the Western diet. As this definition is consistent with previous literature and already cited in the Introduction, we did not add further details to maintain conciseness.

Comments 2: As some of the diet scores (MDS and rMED) were originally designed for non-pregnant adults, their direct application to pregnant women may not fully capture physiological or dietary adaptations during pregnancy. Please justify or discuss the implications of this limitation.

Response 2: Thank you for your valuable comment. We agree with your point that some Mediterranean diet scores, such as the MDS and rMED, were originally designed for non-pregnant adult populations and may not fully reflect pregnancy-specific physiological or dietary adaptations. To address this, we have added an explanation in the Discussion section (paragraph 4) to clarify this point. The revised text now states that although some studies have modified these scores for pregnant women—for example, by excluding alcohol consumption and adjusting cut-off values for food group intakes—such adaptations may still not fully capture pregnancy-specific physiological and nutritional changes, such as increased energy and micronutrient requirements. The revised content appears on page 15, lines 439–453 of the manuscript.

Comments 3: The sentences “Many studies have reported the health benefits of MD, including reduced risks of cardiovascular and total mortality, cancer, type 2 diabetes, and metabolic syndrome,” and “Several studies have been conducted on the Mediterranean region” require appropriate references. Including recent meta-analyses or cohort studies would strengthen the introduction.

Response 3: Thank you for pointing this out. We have added appropriate references to support these statements. Specifically, we cited recent studies demonstrating the health benefits of the Mediterranean diet, including reductions in cardiovascular disease, total mortality, cancer, type 2 diabetes, and metabolic syndrome (Refs. 30–33). We also added references describing studies conducted in Mediterranean regions (Refs. 34,35,38,39). These additions strengthen the background and provide a more comprehensive context for our study.

Comments 4: It would be valuable to know whether any pre-pregnancy dietary or psychological data were available. Could baseline psychological status before MT-1 have influenced the observed associations?

Response 4: Thank you for your valuable comment. We agree with your observation. In the JECS dataset used in this study, dietary information before pregnancy was available, as the MT-1 questionnaire included items regarding pre-pregnancy dietary habits. However, data on pre-pregnancy K6 scores were not available. We recognize that the baseline psychological status before MT-1 could have influenced the observed associations. To address potential confounding, we adjusted for the MT-1 MD score in the analysis. Nevertheless, because data on pre-pregnancy psychological distress were not collected, this remains a limitation of our study. To partially mitigate this limitation, participants with a history of mental disorders were excluded from the analyses. We have added this point to the Limitations section.

Comments 5: Were the same women followed at MT-2, and if so, was there any indication of pre-existing mental vulnerability?

Response 5: Thank you for pointing this out. Yes, the same women were followed from MT-1 to MT-2 in the JECS cohort. To minimize the potential influence of pre-existing mental vulnerability, participants with a history of mental disorders were excluded from the analysis. Therefore, the observed associations are unlikely to be confounded by pre-existing psychiatric conditions. This point has also been added to the Limitations section of the revised manuscript.

Comments 6: Have the authors explored whether using only extreme quartiles (1st vs 4th) for each score would meaningfully change the results? This could highlight direct effects or enhance contrast between groups.

Response 6: Thank you for your valuable comment. We agree with your suggestion and conducted additional analyses by dividing each MD score into equal quartiles and tertiles. However, the number of participants in each group varied considerably, and it was not possible to achieve evenly distributed groups. Therefore, we were unable to proceed further with this analysis. For your reference, we have provided the number of participants for each quartile and tertile classification of the MD scores.

Comments 7: Are there available data linking maternal adherence to MD or psychological distress with neonatal outcomes (e.g., birth weight, gestational age, APGAR score)? Even a brief discussion of these potential associations would enhance the manuscript’s translational value.

Response 7: Thank you for this valuable comment. Data on neonatal outcomes such as birth weight, gestational age, and Apgar score are available in the JECS dataset. However, these outcomes were beyond the scope of the present analysis, which primarily focused on maternal psychological distress during pregnancy. We agree that examining the relationships between maternal adherence to the Mediterranean diet, psychological distress, and neonatal outcomes would provide important insights into potential downstream effects. We have briefly mentioned this point as a future research direction in the Discussion section of the revised manuscript (page 17, line 520–523).

Comments 8: The statement “No significant differences in the proportion of psychological distress were found between the high and low MDS groups (p = 0.08)” requires more cautious interpretation. A p-value of 0.08 may indicate a trend toward significance rather than the absence of an effect. Furthermore, the used statistical test aims to prove that two groups are different, thus, a lack of statistical significance does not mean that two groups are equal. Please revise the wording accordingly and avoid overstating null results.

Response 8: We appreciate your insightful comment. We agree that the previous wording may have implied an absence of any association. Accordingly, we have revised the description in the Results section to adopt a more cautious interpretation. Specifically, we now state that “There was a trend toward a higher proportion of psychological distress in the low MDS group compared with the high MDS group (p = 0.08)” instead of indicating that there were “no significant differences.”

This revision clarifies that the finding suggests a possible trend toward significance rather than the absence of an effect. The corresponding section now reads as follows (page 11, line 310–311).

Comments 9: Between lines 248 and 249, the text appears interrupted. Please restore or clarify the missing portion.

Response 9: Thank you for bringing this to our attention. The interruption in the text was due to an error that occurred during the formatting and layout adjustment of the table and surrounding text. We sincerely apologize for this oversight. The missing portion has now been restored in line 294 to ensure the continuity and accuracy of the text.

Comments 10: Move complete table legends above the respective tables for consistency with journal formatting standards.

Response 10: Thank you for pointing this out. We have modified Table 1[page 5, Table 1, line 211-216] and Table 2 [page 11, Table 2, line 322-323].

Comments 11: Ensure alignment and text formatting are uniform throughout the manuscript, as several unformatted or misaligned passages are present.

Response 11: Thank you for pointing this out. We carefully have checked alignment and text formatting.

Comments 12: Carefully proofread for minor typographical or spacing issues.

Response 12: Thank you for pointing this out. We have carefully and thoroughly proofread the manuscript to correct any minor typographical or spacing errors.

4. Response to Comments on the Quality of English Language

Point: None

5. Additional clarifications

None

Reviewer 3 Report

Comments and Suggestions for Authors

The introduction mentions the Mediterranean diet as one of the main models of healthy eating. I would add that the Mediterranean diet was declared an intangible cultural heritage by UNESCO in 2010 and is still a cornerstone on which dietary recommendations are based for both healthy and sick patients, with the necessary customizations. In the description of the Mediterranean diet, I would also add the importance of consuming fish and extra virgin olive oil, which are not mentioned in the introduction.

Author Response

Response to Reviewer 3 Comments

1. Summary

2. Questions for General Evaluation

Reviewer’s Evaluation

Response and Revisions

Does the introduction provide sufficient background and include all relevant references?

Yes

Are all the cited references relevant to the research?

Yes

Is the research design appropriate?

Yes

Are the methods adequately described?

Yes

Are the results clearly presented?

Yes

Are the conclusions supported by the results?

Yes

3. Point-by-point response to Comments and Suggestions for Authors

Comments 1: The introduction mentions the Mediterranean diet as one of the main models of healthy eating. I would add that the Mediterranean diet was declared an intangible cultural heritage by UNESCO in 2010 and is still a cornerstone on which dietary recommendations are based for both healthy and sick patients, with the necessary customizations. In the description of the Mediterranean diet, I would also add the importance of consuming fish and extra virgin olive oil, which are not mentioned in the introduction.

Response 1: Thank you for pointing this out. We have added a statement indicating that the Mediterranean diet was recognized by UNESCO as an Intangible Cultural Heritage, and we have also emphasized the importance of consuming fish and extra virgin olive oil. Although you mentioned that it was declared in 2010, according to the official UNESCO record, it was inscribed in 2013. We have adopted this official year in the revised manuscript (page 2, lines 94–97).

4. Response to Comments on the Quality of English Language

Point: None

5. Additional clarifications

None

Round 2

Reviewer 1 Report

Comments and Suggestions for Authors

The authors have significantly revised and improved their manuscript.